# Effectiveness of Inactivated COVID-19 Vaccines against Delta-Variant COVID-19: Evidence from an Outbreak in Inner Mongolia Autonomous Region, China

**DOI:** 10.3390/vaccines11020292

**Published:** 2023-01-28

**Authors:** Chao Ma, Chang Huang, Wenrui Wang, Yudan Song, Xiaofeng Jiang, Xiaoling Tian, Boxi Liu, Fuli Chi, Shengli Lang, Dongyan Liu, Weiwei Sun, Lin Tang, Dan Wu, Yifan Song, Junhong Li, Lance Rodewald, Zundong Yin, Zhijie An

**Affiliations:** 1National Immunization Program, Chinese Center for Disease Control and Prevention, Beijing 100050, China; 2Nanning Center for Disease Control and Prevention, Nanning 530023, China; 3Chinese Field Epidemiology Training Program (CFETP), Chinese Center for Disease Control and Prevention, Beijing 100050, China; 4Center for Disease Control and Prevention of Inner Mongolia, Huhehot 010031, China; 5Center for Disease Control and Prevention of Hulunbuir Prefecture, Hulunbuir 021008, China; 6Hexi District Center for Disease Control and Prevention, Tianjin 300211, China

**Keywords:** vaccines effectiveness, inactivated COVID-19 vaccines, outbreak, delta variant, China

## Abstract

Phase 3 clinical trials and real-world effectiveness studies showed that China’s two main inactivated COVID-19 vaccines are very effective against serious illness. In November 2021, an outbreak occurred in the Inner Mongolia Autonomous Region that provided an opportunity to assess the vaccine effectiveness (VE) of these inactivated vaccines against COVID-19 caused by the delta variant. We evaluated VE with a retrospective cohort study of close contacts of infected individuals, using a generalized linear model with binomial distribution and log-link function to estimate risk ratios (RR) and VE. A total of 8842 close contacts were studied. Compared with no vaccination and adjusted for age, presence of comorbidity, and time since last vaccination, full vaccination reduced symptomatic infection by 62%, pneumonia by 64% and severe COVID-19 by 90%; reductions associated with homologous booster doses were 83% for symptomatic infection, 92% for pneumonia and 100% for severe COVID-19. There was no significant decline in two-dose VE for any outcome for up to 325 days following the last dose. There were no differences by vaccine brand. Inactivated vaccines were effective against delta-variant illness, and were highly effective against pneumonia and severe COVID-19; VE was increased by booster doses.

## 1. Introduction

Coronavirus disease 2019 (COVID-19) has remained a global pandemic for the last three years. Following containment in 2020 and during the period of sustained containment, in August 2021, in response to the spread of the highly transmissible delta variant, China adopted a strategy called “dynamic COVID zero.” The essence of this strategy was to take effective and comprehensive measures to rapidly truncate transmission chains and to end outbreaks in a timely manner [1]. Before the SARS-CoV-2 omicron variant became prominent in China, effective implementation of the dynamic COVID zero policy eliminated indigenous transmission of SARS-CoV-2, except for small importation-related outbreaks that were stopped by non-pharmaceutical outbreak responses. These outbreaks provide the best opportunities to measure real-world vaccine effectiveness (VE) of China-produced vaccines on China’s infection-naïve population. Once an importation-related outbreak occurred, the eighth edition of the Protocol for Prevention and Control of COVID-19 in China [2] required contact tracing of SARS-CoV-2 infections and centralized, managed quarantine of close contacts of infected individuals, with systematic reverse transcription-polymerase chain reaction (RT-PCR) testing during quarantine. Managed quarantine of close contacts makes it feasible to conduct cohort studies on vaccine effectiveness (VE).

Two ancestral-strain SARS-CoV-2 whole-virus inactivated COVID-19 vaccines developed and produced in China have been widely used domestically and globally, and are listed by the World Health Organization for emergency use during the pandemic: BIBP-CorV, made by the Beijing Institute of Biological Products (Sinopharm, Beijing China) and CoronaVac, made by Sinovac, Ltd., Beijing, China [3]. As of 22 July 2022, 92.1% of China’s population had received at least one dose of a COVID-19 vaccine, 89.7% had received full primary vaccination, and 71.7% had received a booster dose [4]. By December 31, 2022, over 3.4 billion doses of COVID-19 vaccines had been administered to children, adolescents, and adults in China [5]. Licensure clinical trials conducted overseas during circulation of ancestral strain of SARS-CoV-2 showed both inactivated vaccines to be effective [6,7,8,9]; overseas and domestic real-world studies have shown the vaccines to retain effectiveness against severe COVID-19 caused by the delta and omicron variants [10,11,12,13,14,15].

In China, the dynamic COVID zero policy reduced the number and size of COVID-19 outbreaks. Real-world, domestic evidence of protection with China-produced vaccines has therefore been limited, especially for VE against delta-variant disease. In November 2021, a delta-variant outbreak occurred in the Inner Mongolia Autonomous Region that provided an opportunity to assess primary series and homologous booster dose inactivated vaccine effectiveness against delta-variant-caused illness. We report results of our evaluation and discuss their implications.

## 2. Methods

We used a retrospective cohort study of close contacts of people known to be infected with SARS-CoV-2 during a delta-variant outbreak to estimate primary series and booster dose VE, and the associations between VE, the time since vaccination and the age of the vaccine recipient. Our study took advantage of the public health and social measures used in mainland China during the study period, which included managed quarantine of all close contacts.

### 2.1. Setting and Outbreak

The outbreak started in the city of Manzhouli in the Inner Mongolia Autonomous Region on 28 November 2021, and spread to neighboring Zhalainuoer district in Hulun Buir prefecture, lasting until 16 December 2021. Manzhouli is located in the northeast of the Inner Mongolia Autonomous Region, sharing a border with two countries, Russia and the State of Mongolia, that had continuous community circulation of SARS-CoV-2 throughout 2021. The two aforementioned county-level cities had a combined registered population of 235,000 people; 13.0% < 18 years, 67.5% 18 to 59 years, and 19.5% ≥ 60 years. At the start of the outbreak, two- and three-dose COVID-19 vaccine coverage levels in the general population were 79.4% and 13.4%, respectively. Among 3–17-year-olds, 18–59-year-olds, and ≥60-year-olds, respective two- and three-dose coverage levels were 37.3% and 0.0%, 87.8% and 18.1%, 81.7% and 6.3%. There were 13,200 people who had received one dose but had not completed full primary vaccination by 28 November 2021.

Between 28 November and 16 December 2021, there were 548 COVID-19 cases confirmed in Manzhouli and Zhalainuoer. Among infected individuals, 50.4% were male and 49.6% were female; 47.4% of infections were mild, 46.5% of infected individuals developed pneumonia, 5% of infections were severe, and 1.1% of infections were critical. Eight infections were in children less than three years of age; 26 were in children 3 to 11 years of age; 107 were in children 12 to 17 years of age; 318 were among adults 18 to 59 years of age; and 89 infections were in people 60 years of age or older. Forty-eight infected individuals were unvaccinated, 28 had received one dose, 425 had received two doses, and 47 had received three doses. Of the 500 vaccinated cases, 499 had received inactivated vaccines and one had received an adenovirus type-5 vectored vaccine.

### 2.2. Design and Subjects

We used a retrospective cohort design to determine absolute VE and relative VE (rVE). Close contacts were defined as individuals living in the same residence with an infected person; individuals who had been in the same room, such as a restaurant at the same time with an infected person; and individuals who had been in the same confined space with an infected person up to four days before and after symptom onset of a confirmed case, or four days before and after an RT-PCR positive swab was obtained in an asymptomatically infected person who had not taken effective personal protective measures. We excluded the very few individuals vaccinated with vaccines other than BBIBP-CorV and CoronaVac. Because there were no fully vaccinated 3–11-year-old close contacts, and because there were only 14 (1.5%) unvaccinated 12–17-year-old close contacts (see Appendix A), it was not possible to measure VE for children <18 years of age; these child close contacts were not included in the analytic dataset.

### 2.3. Vaccination Status

Vaccination histories were obtained by interviewing individuals and then assessing vaccination records from national and provincial immunization information systems (IIS). We considered vaccinations to be valid only if they were documented in the national IIS. Based on vaccination history at time of exposure, subjects were categorized into one of four groups: a 0-dose group (those did not receive any COVID-19 vaccine), a 1-dose group (those who received only one dose), a full 2-dose group (those who received two doses of vaccine >14 days before SARS-CoV-2 exposure, subdivided into 15–90, 91–180, and 181 up to 325 days before exposure), and a 3-dose group (those who received 3 doses of vaccine, subdivided into 0–6 or ≥7 days between the third dose and exposure). A second dose was considered valid only when given at least 14 days after the first dose; doses with shorter time intervals were considered invalid and not counted in vaccination status assessment.

### 2.4. Clinical Outcomes

We evaluated VE for protection from three clinical outcomes: symptomatic COVID-19, COVID-19 pneumonia, and severe COVID-19. Case classifications were based on the COVID-19 Diagnosis and Treatment Protocol (trial eighth edition) and the COVID-19 Prevention and Control Protocol (eighth edition) categories of asymptomatic, mild, moderate, severe, and critically severe infection. A symptomatic COVID-19 outcome included all mild, moderate, severe, and critically severe cases. A COVID-19 pneumonia outcome included moderate, severe, and critically severe cases with evidence of pneumonia. A severe illness outcome included severe and critically severe cases. There were no fatal cases.

Symptomatic COVID-19 was defined as SARS-CoV-2 infection with any symptom or sign of COVID-19, regardless of severity. COVID-19 pneumonia was diagnosed based on clinical symptoms and computed tomographic (CT) imaging (chest CT imaging findings consistent with COVID-19 pneumonia were bilateral ground-glass opacities with predominantly peripheral and lower lobe distribution). Severe illness met any of the following criteria: (a) respiratory distress, or a respiratory rate of 30 breaths/minute or more; (b) a resting oxygen saturation of less than 93%; (c) a ratio of arterial partial pressure of oxygen to fractional inspired oxygen concentration of <300 mmHg (with adjustment for altitude above 1000 m); or (d) patients with pneumonia having >50% lesion progression within 24–48 h on chest imaging. Critical illness met any of the following criteria: (a) respiratory failure or need for mechanical ventilation; (b) shock; or (c) other organ failure that required monitoring and treatment in an intensive care unit [7].

### 2.5. Statistical Analyses

VE was determined in this cohort of quarantined close contacts by comparing the vaccination status of quarantined individuals who ultimately tested positive for SARS-CoV-2 with those who remained RT-PCR negative. To calculate unadjusted VE, the relative risk (RR) of each outcome was calculated in reference to the 0-dose group using the equation VE = (1 − RR) × 100. We calculated 3-dose relative VE (rVE) in reference to the 2-dose group. For adjusted VE (aVE) and rVE, age, presence of comorbidities, interval between most recent dose and SARS-CoV-2 exposure were considered potentially confounding variables in multivariate analyses. We used a generalized linear model with a binomial distribution and log-link function to calculate adjusted risk ratios (aRR), which were used to calculate aVE and adjusted rVE. Cox proportional hazards models were used to create survival curves adjusted for age, presence of comorbidities, interval between exposure and most recent vaccination date; RR values were tested for significance with a Chi-square test. We used R (version 4.1.2) for drawing charts, statistical analysis, generalized linear model analysis, and Cox proportional hazards analysis. *p*-values < 0.05 (two-tailed) were considered statistically significant.

### 2.6. Ethical Considerations

COVID-19 is a Level 2 infectious disease being managed as a Level 1 (highest) infectious disease during the study period. Investigation of outbreaks and assessment of vaccine effectiveness are required activities of disease prevention and control units. Our study used routinely collected data required by the Protocol for Prevention and Control of COVID-19. As an analysis of routinely collected data, the study was exempt from Ethical Review Committee review. Individual-identifying information was not retained in analytic data sets, and informed consent for this retrospective study was not required or obtained.

## 3. Results

### 3.1. Characteristics of Subjects

Table 1 shows the characteristics of the study subjects, and Appendix A shows the disposition of all close contacts. Among 10,788 close contacts, 8842 met the inclusion criteria (≥18 years old and not vaccinated with a COVID-19 vaccine other than an inactivated vaccine) and were included in the VE analyses. Among the included close contacts, 4451 (50.3%) were male, 7722 (87.3%) were 18–59 years old, and 1120 (12.7%) were ≥60 years old. Among the close contacts, 373 (4.2%) tested positive by RT-PCR for SARS-CoV-2 during quarantine, and the remaining 8469 (95.8%) tested negative throughout quarantine; 1551 (17.5%) close contacts self-reported one or more comorbidities, 5539 (62.6%) reported no comorbidities, and presence of a comorbidity was unknown for the remaining 1752 (19.8%). The top five comorbidities reported by close contacts were hypertension (8.5%), heart disease (4.9%), diabetes (3.2%), respiratory disease (2.1%), and cerebrovascular disease (1.5%).

A total of 403 (4.56%) close contacts were unvaccinated; 100 (1.13%) had received one dose; 6396 (72.33%) had received two doses; and 1943 (21.97%) had received three doses (two primary series doses and a booster dose). Medians (interquartile intervals in days) from the last dose prior to SARS-CoV-2 exposure were 93.5 days (IQR:38–178 days) for 1-dose subjects, 171 days (IQR:160–188 days) for two-dose subjects, and 15 days (IQR:8–33 days) for 3-dose subjects (Figure 1A).

All 373 SARS-CoV-2-infected close contacts met the criteria for symptomatic illness. Among infected individuals, 244 developed pneumonia, 31 developed severe COVID-19 (Table 1), and none died; 288 were 18–59 years, and 85 were ≥60 years. There were no severe or severe/critical cases among the booster dose recipients. Adjusted Cox regression survival curves for each clinical outcome are shown by vaccination status in Figure 1B1–3. For symptomatic, pneumonia, and severe COVID-19 outcomes, the outcome probability was highest in the unvaccinated group, followed in order by the 1-dose, 2-dose and 3-dose groups.

### 3.2. Vaccine Effectiveness

Figure 2, Figure 3 and Figure 4 show aVE and rVE by vaccination status, age group, and time since vaccination. Among subjects ≥18 years old, adjusted VE levels for 1-dose, 2-dose, and booster vaccination were 45.99% (95%CI: −31.99–77.90), 61.67% (95%CI: 46.06–72.77) and 82.54% (95%CI: 71.59–89.28) against symptomatic COVID-19; 39.39% (95%CI: −61.99–77.32), 64.00% (95%CI: 48.13–75.02), and 91.55% (95%CI: 81.63–96.12) against pneumonia; and 100% (95%CI: −Inf–100), 89.79% (95%CI: 70.47–96.47), and 100% (95%CI: −Inf–100) against severe/critical COVID-19, respectively. Relative to the 2-dose group, adjusted rVEs of the booster dose against the three clinical outcomes were 56.37% (95%CI: 40.24–68.15), 80.15% (95%CI: 65.35–88.63), and 100% (95%CI: −Inf–100), respectively. Adjusted VEs were higher among ≥60-year-olds than 18–59-year-olds. There was no significant decline in aVE with the number of days since vaccination in the 2-dose group. In the booster-dose group, aVEs of 86.08% (95%CI: 68.66–93.82) for symptomatic, 96.89% (95%CI: 76.93–99.58) for COVID-19 pneumonia, and 100% (95%CI: −Inf–100) for severe COVID-19 were observed within 7 days following administration of the third dose.

Figure 5 shows VE by vaccine brand against the three clinical outcomes. Among the 8439 individuals vaccinated with an inactivated vaccine, 1567 received homologous CoronaVac, 1585 received homologous BBIBP-CorV, and 5300 received a dose of BIBP-CorV followed by CoronaVac or vice versa (combined). For symptomatic COVID-19, aVEs for BBIBP-CorV, CoronaVac, and combined were 64.93% (95%CI: 40.94–79.18), 61.46% (95%CI: 37.62–76.19) and 63.32% (95%CI: 44.73–75.66) in the 2-dose group and 80.71% (95%CI: 64.6–89.49), 84.07% (95%CI: 52.09–94.71), and 80.34% (95%CI: 63.36–89.45) in the booster-dose group. Against pneumonia, aVEs were 70.26% (95%CI: 45.36–83.82), 72.50% (95%CI: 51.01–84.56), 65.76% (95%CI: 45.73–78.40) in the 2-dose group and 89.29% (95%CI: 69.12–96.28), 81.81% (95%CI: 7.59–96.42), and 88.24% (95%CI: 63.71–96.19) for the booster-dose group. There were no statistically significant differences in aVE by vaccine brand.

## 4. Discussion

Our real-world evaluation of the performance of China’s inactivated COVID-19 vaccines during an outbreak showed that these vaccines provided high-level protection against COVID-19 pneumonia and severe COVID-19 caused by the delta variant, similar to vaccine effectiveness levels against illnesses caused by ancestral and previous SARS-CoV-2 strains. Compared with no vaccination and adjusted for age, presence of a comorbidity, and time since vaccination, full primary vaccination reduced symptomatic COVID-19 by 62%, pneumonia by 64%, and severe COVID-19 by 90%; booster-dose vaccination reduced symptomatic COVID-19 by 83%, pneumonia by 92%, and severe COVID-19 by 100%. Adjusted vaccine effectiveness levels were higher among individuals 60 years or older than among 18–59-year-olds. For up to 325 days following the last dose of vaccine, there was no significant decline in 2-dose vaccine effectiveness against any clinical outcome. Vaccine effectiveness did not vary by brand of vaccine. Continued use of these inactivated COVID-19 vaccines is warranted, and the use of booster doses is necessary for optimal protection from severe forms of COVID-19.

The study was conducted during full implementation of the dynamic COVID zero policy in mainland China. During this time, there were relatively very few cumulative SARS-CoV-2 infections, implying that the study population was infection-naïve up to the start of the outbreak and therefore lacked hybrid immunity. As a consequence, the vaccine effectiveness estimates are of only vaccine-induced protection, without the influence of prior infection.

COVID-19 vaccines made by other technical platforms have similar performance characteristics, showing greatest effectiveness against the more serious forms of delta-variant COVID-19 and requiring booster doses to retain or augment effectiveness [16]. The International Vaccine Access Center of Johns Hopkins Bloomberg School of Public Health and the World Health Organization (WHO) continuously monitor vaccine effectiveness studies of WHO-emergency-listed COVID-19 vaccines, serving as an invaluable reference for the most important vaccine effectiveness performance characteristics of the vaccines being used to mitigate the pandemic [17].

Our results are consistent with other domestic vaccine effectiveness studies of inactivated and other China-produced vaccines against the delta and omicron variants. Earlier in 2021, Ma and colleagues used a delta-variant outbreak in Ruili City of Yunnan Province to assess vaccine effectiveness, finding protection afforded by inactivated and adenovirus 5-vectored vaccines to be 68% to 77% against pneumonia and 100% against severe COVID-19, with no deaths among any subjects [18]. Kang and colleagues found similar protection against delta in a Guangdong Province outbreak: there was 78% VE against pneumonia and 100% VE against severe/critical COVID-19 [19]. A study of delta and omicron outbreaks in Henan Province in early 2022 found that a homologous inactivated vaccine booster dose reduced delta COVID-19 pneumonia 82% relative to primary series, and that primary series alone reduced omicron pneumonia by 66% [20]. McMenamin and colleagues found that in Hong Kong during an Omicron BA.2 wave, the homologous-boosted inactivated vaccine was 98% effective in preventing severe/fatal COVID-19; this is on par with the homologous boosted mRNA vaccine [10]. Without boosting, inactivated primary series vaccination was 70% effective, showing the necessity of a booster dose. Against symptomatic infection, inactivated vaccines and mRNA vaccines had similar vaccine effectiveness levels: 25% and 35%, respectively, showing that neither vaccine affords high protection from infection. The Hong Kong study also showed three-dose inactivated vaccine effectiveness against severe/fatal COVID-19 was very high among the elderly: 97%, 95%, and 97% for people in their 60s, 70s, and 80s or older. In the early 2022 Shanghai Omicron BA.2 outbreak, Huang and colleagues had findings similar to those in Hong Kong: three-dose inactivated VE among adults was modest against infection but was 93% against severe COVID-19, and 96% against fatal COVID-19 [21].

Our study has program implications. The first implication is that the COVID-19 vaccination campaign should continue, as the vaccines are having measurably strong effectiveness at preventing serious forms of COVID-19. It is the job of the immunization program to use vaccines to protect the population from serious and fatal COVID-19, and the findings from this study show that the vaccines not only provide protection, but that this protection has a reasonable duration of at least 325 days against the delta variant. The second program implication is that booster doses are important for attaining maximal protection, and therefore, the booster dose campaign should continue, attempting to reach all booster dose-eligible individuals. A third program implication is that monitoring the effectiveness and safety of vaccines is essential in the COVID-19 pandemic. Every opportunity must be taken to determine whether the vaccines being used retain effectiveness. The best opportunities in mainland China are in outbreaks such as this one in Inner Mongolia, because outbreaks are generally caused by a single variant, have well-defined exposed populations, and occur over discrete intervals. Variants have been emerging into global prominence at a rapid pace because SARS-CoV-2 is a novel virus, evolving as it traverses the world’s population with a variety of immune selection pressures. The fourth implication is that monitoring the severity of breakthrough infections is critically important. During the dynamic COVID zero policy, the number of infections was small. However, upon lifting the dynamic COVID zero policy, the virus spread rapidly due to limited effectiveness against infection and transmission. Under conditions of a high force of infection, the healthcare system may be overly stressed if the proportion of severe and critical cases is too high. The results of this study show that the proportion of serious/critical infections is low. This information is useful for mathematical modeling for predicting stress on the healthcare system upon lifting the dynamic COVID zero policy. Fifth, the results of this study can be used in communications to help assure the public that the vaccines produced and used in China are working well and providing protection from serious illness, and that the vaccination campaign is providing measurable benefit.

The strengths of our study include (1) a study population of individuals with known exposure to delta-variant SARS-CoV-2 who are closely monitored and systematically tested for infection, and ascertainment of vaccination status of the subjects; (2) ascertainment of vaccination status with medical records and documentation, indexed by subjects’ national identification numbers; (3) the study being conducted during the dynamic COVID zero policy in an infection-naïve population, so that the vaccine effectiveness findings represent only vaccine-induced protection, without influence of hybrid immunity; and (4) the ability to estimate both absolute vaccine effectiveness and relative vaccine effectiveness. The unvaccinated group allowed estimation of absolute vaccine effectiveness, with relative VE showing the added protection afforded by the booster dose. When vaccination coverage levels become extremely high, the unvaccinated group will become increasingly different from vaccinated individuals, and different methods may be needed to assess vaccine effectiveness. Thus, the timing of this study makes it possible to estimate absolute vaccine effectiveness.

There are several limitations to our study. One limitation is the relatively small size of the outbreak due to application of public health and social measures to stop the outbreak that were in effect during the study period. The small size of the outbreak limited the number of exposed close contacts, which in turn precluded determination of some useful subgroup analyses such as VE by comorbidity. Although we were able to adjust for the presence of one or more comorbidities, we could not determine VE by any specific comorbidity. A second limitation is that we were only able to determine the VE of the inactivated COVID-19 vaccines in common use in China. Due to limited use of non-inactivated COVID-19 vaccines in this population, we were unable to determine VE for other approved COVID-19 vaccines. Third, our study was observational, and therefore reports associations rather than causal inferences. There may have been unmeasured confounding variables that influenced the vaccine effectiveness results. However, observational studies are the most commonly used studies for assessing vaccine effectiveness under real-world conditions, and observational studies do not raise ethical issues of random allocation to vaccine or placebo when the efficacy of the vaccine is known and effective vaccines are widely available. A fourth limitation is that because the booster dose policy during the study period was limited to homologous booster doses, we were unable to determine the differential effectiveness of heterologous versus homologous booster doses following primary series vaccination with inactivated COVID-19 vaccines.

In conclusion, the BIBP-CorV and SinoVac inactivated COVID-19 vaccines were highly effective against the delta variant, were most effective against the more severe outcomes and when boosted, and were not associated with a decline in effectiveness up to 325 days. The results of this study add to the growing confidence that population immunity will be established with these vaccines and boosted with the same or other COVID-19 vaccines, and the growing confidence that when used as recommended, these vaccines can greatly decrease severe outcomes from SARS-CoV-2 infection. During the time of China’s dynamic COVID zero policy, it is critically important to attain as high full-series and booster-dose coverage as possible for vaccine-eligible people who do not have valid contraindications, especially the elderly, as they suffer disproportionately from COVID-19. All modest-size or larger COVID-19 outbreaks should be used to evaluate population immunity against circulating SARS-CoV-2 variants.

## Figures and Tables

**Figure 1 vaccines-11-00292-f001:**
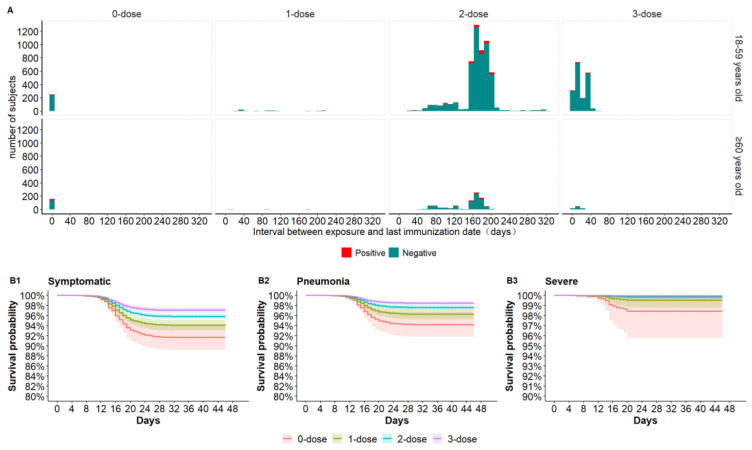
Interval (days) between exposure and the last dose of vaccine and probability of each clinical outcome by vaccination group based on Cox proportional hazards analyses. (**A**) Distribution of interval between exposure and last dose of all subjects by age group and dose number. (**B1**–**B3**) Outcome probability by vaccine dose number for symptomatic disease, pneumonia, and severe COVID-19, adjusting for age, presence of a comorbidity, and interval between exposure and last dose.

**Figure 2 vaccines-11-00292-f002:**
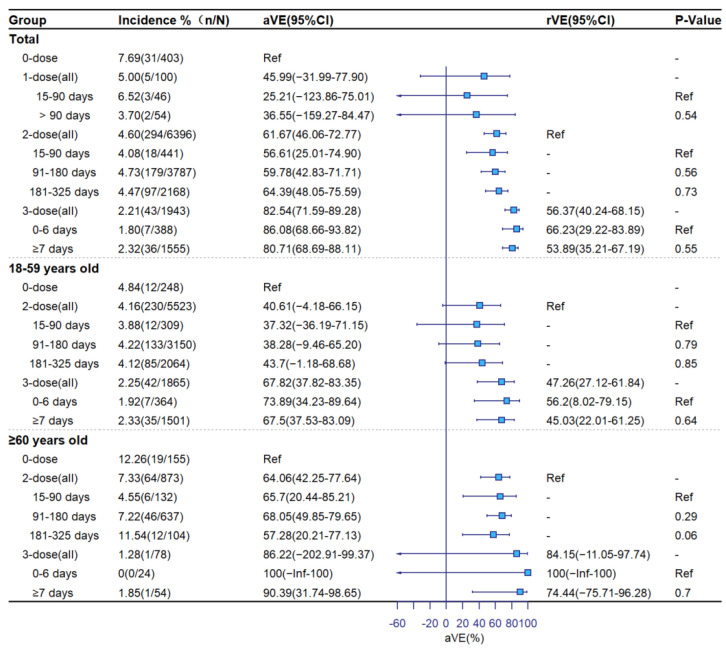
Adjusted and relative vaccine effectiveness against symptomatic COVID-19.

**Figure 3 vaccines-11-00292-f003:**
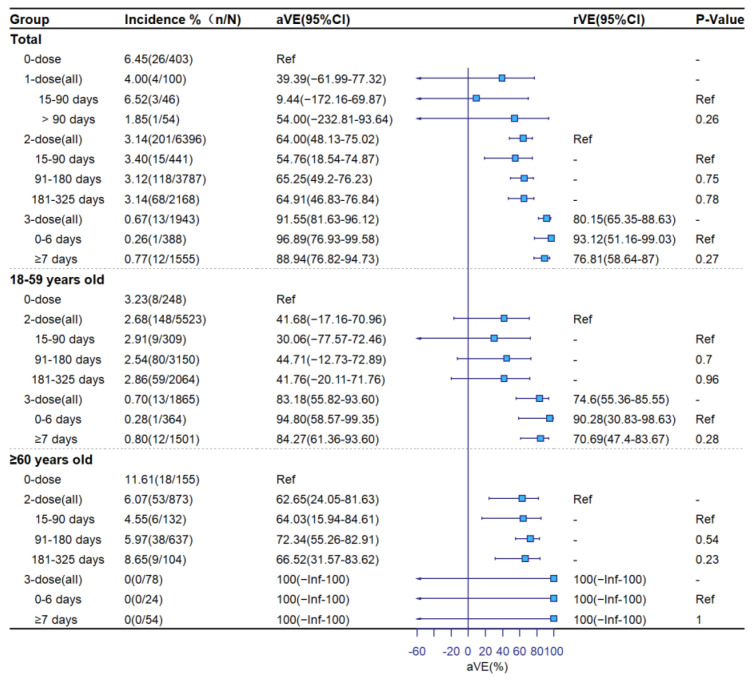
Adjusted and relative vaccine effectiveness against pneumonia.

**Figure 4 vaccines-11-00292-f004:**
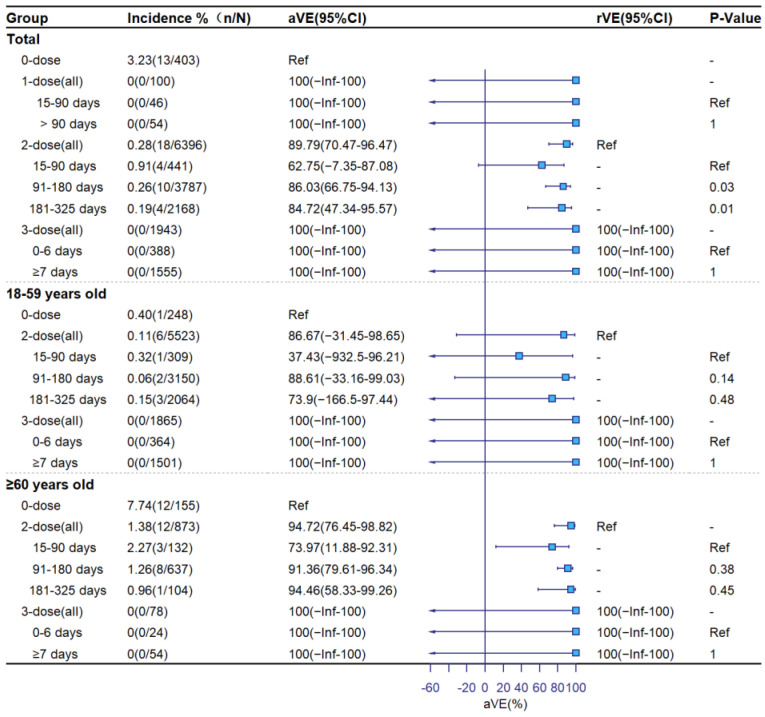
Adjusted and relative vaccine effectiveness against severe COVID-19.

**Figure 5 vaccines-11-00292-f005:**
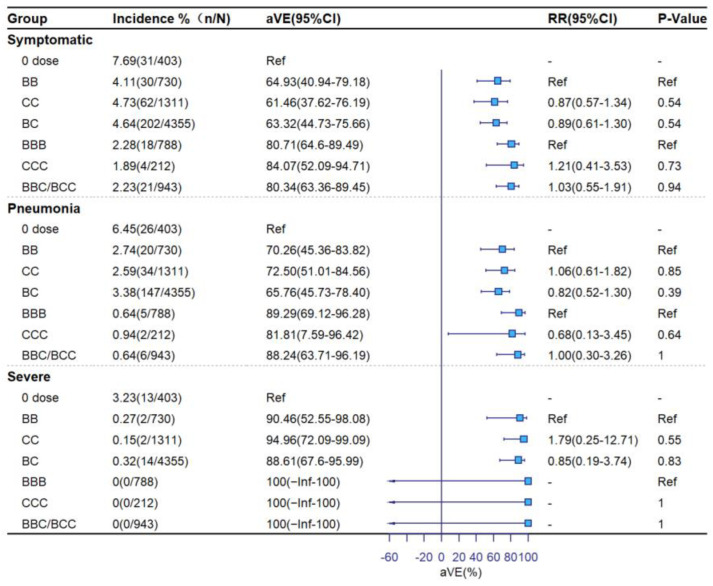
Adjusted vaccine effectiveness by vaccine manufacturer (BB: both BBIBP-CorV for 2 doses, CC: both CoronaVac for 2 doses, BC: one dose of each brand for 2 doses, BBB: both BBIBP-CorV for 3 doses, CCC: both CoronaVac for 3 doses, BCC/BBC: at least one dose of each brand vaccine for 3 doses).

**Table 1 vaccines-11-00292-t001:** Characteristics of close contact study subjects by vaccination status.

Characteristic	Number (%)	Number of Doses (%)				*p*
		0	1	2	3	
Total	8842 (100%)	403 (4.56)	100 (1.13)	6396 (72.33)	1943 (21.97)	
Gender						<0.001
Male	4451 (50.3)	172 (42.7)	62 (62)	3182 (49.7)	1035 (53.3)	
Female	4391 (49.7)	231 (57.3)	38 (38)	3214 (50.3)	908 (46.7)	
Age(years)						<0.001
18–59	7722 (87.3)	248 (61.5)	86 (86)	5523 (86.4)	1865 (96)	
≥60	1120 (12.7)	155 (38.5)	14 (14)	873 (13.6)	78 (4)	
Symptomatic						<0.001
Yes	373 (4.2)	31 (7.7)	5 (5)	294 (4.6)	43 (2.2)	
No	8469 (95.8)	372 (92.3)	95 (95)	6102 (95.4)	1900 (97.8)	
Pneumonia						<0.001
Yes	244 (2.8)	26 (6.5)	4 (4)	201 (3.1)	13 (0.7)	
No	8598 (97.2)	377 (93.5)	96 (96)	6195 (96.9)	1930 (99.3)	
Severe case						<0.001
Yes	31 (0.4)	13 (3.2)	0 (0)	18 (0.3)	0 (0)	
No	8811 (99.6)	390 (96.8)	100 (100)	6378 (99.7)	1943 (100)	
Comorbidity						<0.001
Yes	1551 (17.5)	129 (32)	20 (20)	1091 (17.1)	311 (16)	
No	5539 (62.6)	113 (28)	50 (50)	4040 (63.2)	1336 (68.8)	
Unknown	1752 (19.8)	161 (40)	30 (30)	1265 (19.8)	296 (15.2)	

## Data Availability

Not applicable.

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
