# Peer review of "Effectiveness of Inactivated COVID-19 Vaccines against Delta-Variant COVID-19: Evidence from an Outbreak in Inner Mongolia Autonomous Region, China"

_vaccines, 2023, doi:10.3390/vaccines11020292_

Round 1
Reviewer 1 Report
This retrospective cohort study aimed to assess vaccine effectiveness of China-produced inactivated vaccines against COVID-19 caused by Delta variants in Inner Mongolia Autonomous Region. The topics is of interest and i would like to congratulate with Authors for their effort
My comments are:
1) Add in the discussion section a word on the importance of animal serum proteomic in order to identify novel biomarkers for animal welfare, early diagnosis, prognosis and monitoring of infectious disease and treatment (as develop of new vaccines).
[Di Girolamo, Amato AD', Lante I, Signore F, Muraca M, Putignani L. Farm animal
299 serum proteomics and impact on human health. Internat J Molec Sci 2014;15, 15396-15411. 300 doi:10.3390/jims150915396.]
Add in the discussion section a citation about vaccines and pregnancy [Cavaliere, A.F.; Zaami, S. Flu and Tdap Maternal Immunization Hesitancy in Times of COVID-19: An Italian Survey on Multiethnic Sample. Vaccines 2021, 9, 1107.] [WAPM (World Association of Perinatal Medicine) Working Group on COVID-19. Maternal and perinatal outcomes of pregnantwomen with SARS-CoV-2 infection. Ultrasound Obstet. Gynecol. 2021, 57, 232–241]; [Di Mascio D, Sen C, Saccone G, Galindo A, Grünebaum A, Yoshimatsu J, et al. Risk factors associated with adverse fetal outcomes in pregnancies affected by Coronavirus disease 2019 (COVID-19): a secondary analysis of the WAPM study on COVID-19. J Perinat Med. 2020 Nov 26;48(9):950-958. doi: 10.1515/jpm-2020-0355. Erratum in: J Perinat Med. 2020 Dec 02;49(1):111-115] and description of transplacental passage of specific SARS-CoV-2 IgG from mothers who contracted natural infection to their newborns [Marchi, L.; Vidiri, A. SARS-CoV-2 IgG “heritage” in newborn: A credit of maternal natural infection. J. Med. Virol. 2022;]
Author Response
Response to Reviewer 1 Comments
This retrospective cohort study aimed to assess vaccine effectiveness of China-produced inactivated vaccines against COVID-19 caused by Delta variants in Inner Mongolia Autonomous Region. The topics is of interest and i would like to congratulate with Authors for their effort
My comments are:
1) Add in the discussion section a word on the importance of animal serum proteomic in order to identify novel biomarkers for animal welfare, early diagnosis, prognosis and monitoring of infectious disease and treatment (as develop of new vaccines).
[Di Girolamo, Amato AD', Lante I, Signore F, Muraca M, Putignani L. Farm animal
299 serum proteomics and impact on human health. Internat J Molec Sci 2014;15, 15396-15411. 300 doi:10.3390/jims150915396.]
Add in the discussion section a citation about vaccines and pregnancy [Cavaliere, A.F.; Zaami, S. Flu and Tdap Maternal Immunization Hesitancy in Times of COVID-19: An Italian Survey on Multiethnic Sample. Vaccines 2021, 9, 1107.] [WAPM (World Association of Perinatal Medicine) Working Group on COVID-19. Maternal and perinatal outcomes of pregnant women with SARS-CoV-2 infection. Ultrasound Obstet. Gynecol. 2021, 57, 232–241]; [Di Mascio D, Sen C, Saccone G, Galindo A, Grünebaum A, Yoshimatsu J, et al. Risk factors associated with adverse fetal outcomes in pregnancies affected by Coronavirus disease 2019 (COVID-19): a secondary analysis of the WAPM study on COVID-19. J Perinat Med. 2020 Nov 26;48(9):950-958. doi: 10.1515/jpm-2020-0355. Erratum in: J Perinat Med. 2020 Dec 02;49(1):111-115] and description of transplacental passage of specific SARS-CoV-2 IgG from mothers who contracted natural infection to their newborns [Marchi, L.; Vidiri, A. SARS-CoV-2 IgG “heritage” in newborn: A credit of maternal natural infection. J. Med. Virol. 2022;]
Response: We appreciate the positive comment on our study.
We have reviewed these recommended articles, but it is not clear that they would contribute to the discussion in our manuscript.
The first article requested to cite (Farm Animal Serum Proteomics and Impact on Human Health) is an article on animal proteomics that seeks to increase use of proteomics studies to support animal husbandry to improve animal and human health. Although animal proteomics is important, it doesn’t seem relevant to our human vaccine effectiveness study. Our preference is to not cite this article.
The second article requested to cite (Flu and Tdap Maternal Immunization Hesitancy in Times of COVID-19: An Italian Survey on Multiethnic Sample) is about vaccine hesitancy for maternal vaccination. Since our study did not address pregnancy in any of the subjects, and since pregnancy was a contraindication to COVID-19 vaccination in China at the time of the study (and is to this day), the relevance of a pregnancy vaccine hesitancy study to our discussion is not apparent. Our preference is to not cite this manuscript.
The third (Maternal and perinatal outcomes of pregnant women with SARS-CoV-2 infection) and fourth (Risk factors associated with adverse fetal outcomes in pregnancies affected by Coronavirus disease 2019 (COVID-19): a secondary analysis of the WAPM study on COVID-19) articles requested to cite are about adverse birth outcomes from SARS-CoV-2 infection. Since we did not evaluate any birth outcomes of any subjects, these two articles do not seem relevant to our study. Our preference is to not cite these two articles.
The fifth and last article (SARS-CoV-2 IgG “heritage” in newborn: A credit of maternal natural infection) requested to cite is about transplacental transfer of antibody to the fetus following maternal SARS-CoV-2 infection. Our study did not address immunogenicity, maternal infection, or maternal vaccination, so the relevance of this study to our study is not apparent. Our preference is to not cite this study.
Reviewer 2 Report
Dear Authors, please take into your consideration the following comments.
Please, limit the Abstract to 200 words following Journal’s guidelines. Also, create an unstructured Abstract without the words “Background”, “Methods”, etc.
Please update your data regarding the number of doses that had been administered to children, adolescents, and adults in China. You mention data until May 29, 2022.
You report that a 548-case Delta-variant outbreak occurred in November (Abstract section) and December (Introduction section). Please clarify.
Clarify the aim of your study in the last paragraph of Introduction without details, e.g. the number of cases or the country.
Please, expand the Introduction section providing more details regarding the effectiveness of inactivated COVID-19 vaccines against Delta-2 variant, factors that associate with effectiveness, the percentage of people in China that are unvaccinated, partially vaccinated, fully vaccinated, etc.
Please, provide a Table to summarize the numbers that you report in Section 2.1.
I believe that it would be better to remove from the multivariate analysis the variables with zero cases, e.g. >60 years old with 3-doses. RRs and CIs in these cases are of limited value.
Please, in the Discussion section, compare the effectiveness of inactivated COVID-19 vaccines against Delta-2 variant in your study with non-inactivated COVID-19 vaccines in similar studies.
I believe that there are more limitations to your study. For example, you did not collect information regarding preventive measures (e.g. mask wearing) that participants used. Thus, confounding factors could still be a bias in your study.
Author Response
Response to Reviewer 2 Comments
Point 1: Please, limit the Abstract to 200 words following Journal’s guidelines. Also, create an unstructured Abstract without the words “Background”, “Methods”, etc.
Response1: We have done so. Now the abstract is unstructured and less than 200 words.
Point 2: Please update your data regarding the number of doses that had been administered to children, adolescents, and adults in China. You mention data until May 29, 2022.
Response2:We have done so. COVID-19 vaccination data have been updated in the manuscript to December 31, 2022, which includes the study period.
Point 3: You report that a 548-case Delta-variant outbreak occurred in November (Abstract section) and December (Introduction section). Please clarify.
Response3:We appreciate the reviewer for picking up this discrepancy. The outbreak occurred in late November 2021. We have revised the Introduction so that it is consistent.
Point 4: Clarify the aim of your study in the last paragraph of Introduction without details, e.g. the number of cases or the country.
Response4:We removed nonessential details in the last paragraph of the Introduction, which states the purpose of the study. We removed “548 case”, but we kept “In China” at the beginning of this paragraph, because we are introducing the national dynamic COVID-zero policy of the country and during the study period, implementation of this policy has kept the population infection-naïve, and therefore reliant on only vaccine-induced immunity.
The purpose of the study is now stated as, “In November 2021, a Delta-variant outbreak occurred in Inner Mongolia that provided an opportunity to assess primary series and homologous booster dose inactivated vaccine effectiveness against Delta-variant-caused illness.”
Point 5: Please, expand the Introduction section providing more details regarding the effectiveness of inactivated COVID-19 vaccines against Delta-2 variant, factors that associate with effectiveness, the percentage of people in China that are unvaccinated, partially vaccinated, fully vaccinated, etc.
Response5:We appreciate the suggestion and agree with both points. The manuscript needs to describe the results of this study in the context of other studies of VE against Delta. To address inactivated VE against Delta variant infection, we cite references 14 through 19. All of these studies report VE against Delta. Rather than going into detail in the Introduction section, our preference is to go into this detail in the Discussion section – in a paragraph that compares our study results with other study results.
To address the point about coverage, we added “By 22 July 2022, 3.417 billion doses of COVID-19 vaccines had been administered to children, adolescents, and adults in China; 92.1% received at least one dose, 89.7% received full primary vaccination, and 71.7% received a booster dose” in the Introduction.
Point 6: Please, provide a Table to summarize the numbers that you report in Section 2.1.
Response6:We put data in section 2.1 in tables below. However, we did not put these table inside the text, since we have already had six tables & figures. We believe that in the text, the data are clearly described, and that including these data will adequately address the reviewer’s good comment.
The vaccination status of the 548 cases in the outbreak
|
Age group |
unvaccination |
1-dose |
2-dose |
3-dose |
total |
|
<3 |
8 |
0 |
0 |
0 |
8 |
|
3-11 |
7 |
19 |
0 |
0 |
26 |
|
12-17 |
1 |
2 |
104 |
0 |
107 |
|
18-59 |
12 |
6 |
254 |
46 |
318 |
|
≥60 |
20 |
1 |
67 |
1 |
89 |
|
total |
48 |
28 |
425 |
47 |
548 |
Point 7: I believe that it would be better to remove from the multivariate analysis the variables with zero cases, e.g. >60 years old with 3-doses. RRs and CIs in these cases are of limited value.
Response7:Thanks for the positive suggestions. We discussed on this with co-authors. We think that no cases occurred in 3-dose recipients also provides important information when comparing to the unvaccinated reference group, which does have cases. That there were no cases in the 3-dose recipients and that the unvaccinated group had cases is what one would expect to see in a highly effective vaccine. Our preference is to keep this important information in the multivariate analyses.
Point 8: Please, in the Discussion section, compare the effectiveness of inactivated COVID-19 vaccines against Delta-2 variant in your study with non-inactivated COVID-19 vaccines in similar studies.
Response8:We agree with the reviewer that comparison is important. In the fourth paragraph of the Discussion, we state, “earlier in 2021, Ma and colleagues use a Delta-variant outbreak in Ruili city of Yunnan province to assess VE, finding protection afforded by inactivated and adenovirus 5-vectored vaccines to be 68% to 77% against pneumonia and 100% against severe COVID-19, with no deaths among any subjects”. This is the only study in mainland China that evaluated VE against Delta that includes both inactivated and non-inactivated COVID-19 vaccines.
Because one of our results is on duration of protection (i.e., no loss of protection for 325 days, as highlighted in the abstract and first paragraph of the Discussion), in the third paragraph of the Discussion, we cite the Feikin review of VE studies on duration of protection. Feikin et al include the Delta variant in their analyses. The sentence in the Discussion section states, “COVID-19 vaccines made by other technical platforms have similar performance characteristics, showing greatest effectiveness against the more serious forms of Del-ta-variant COVID-19 and requiring booster doses to retain or augment effectiveness.” In the second half of that paragraph, we refer the interested readers to the ongoing review of VE studies that is conducted by Patel and Feikin at US CDC and WHO.
Point 9: I believe that there are more limitations to your study. For example, you did not collect information regarding preventive measures (e.g. mask wearing) that participants used. Thus, confounding factors could still be a bias in your study.
Response9:We agree with the your good point. Every observational study can have unmeasured confounding factors. For preventive measures, we did our best to reduce potential confounding. We obtained information about close contacts under the same prevention and control measures. Close contacts were defined as individuals living in the same residence with an infected person; being in the same room, such as in a restaurant at the same time with an infected person; being in the same confined space with an infected person up to four days before and after symptom onset of a confirmed case or four days before and after an RT-PCR positive swab was obtained in an asymptomatically-infected person who failure to take effective personal protection measures. We believe that the preventive measures are about as equivalent as they can be for the close contacts. These points are stated in the Methods section of the manuscript.
To address the point that there can be unmeasured confounding variables, we included as a limitation that “Third, our study was observational, and therefore reports associations rather than causal inferences.”
Reviewer 3 Report
The article entitled “Effectiveness of Inactivated COVID-19 Vaccines against Delta-variant COVID-19: Evidence from an Outbreak in Inner Mongolia Autonomous Region, China” is a retrospective study focusing on the performane of the inactivated COVID-19 vaccines produced by Sinovac and SinoPharm about the COVID-19 pneumonia and severe COVID-19. The study design is well-projected and the aim is well-oriented. The results are well-described and show interesting data. I have only one observation about the comorbidities. I think that it would be important to list the comorbidities from used database and possibly to report wether or no the vaccine affected the comordidity. Therefore, I think that this article is publishable after adding of this information.
Author Response
Response to Reviewer 3 Comments
The article entitled “Effectiveness of Inactivated COVID-19 Vaccines against Delta-variant COVID-19: Evidence from an Outbreak in Inner Mongolia Autonomous Region, China” is a retrospective study focusing on the performane of the inactivated COVID-19 vaccines produced by Sinovac and SinoPharm about the COVID-19 pneumonia and severe COVID-19. The study design is well-projected and the aim is well-oriented. The results are well-described and show interesting data. I have only one observation about the comorbidities. I think that it would be important to list the comorbidities from used database and possibly to report wether or no the vaccine affected the comordidity. Therefore, I think that this article is publishable after adding of this information.
Response:We appreciate the positive comments and we agree with the point about comorbidities.
To address the your request, we now include the top five comorbidities. The Results section states, “The top five comorbidities reported by close contacts were hypertension (8.5%), heart dis-ease (4.9%), diabetes (3.2%), respiratory disease (2.1%), and cerebrovascular disease (1.5%).”
We also state in the Limitations section that a “limitation of our study is that the relatively small size of the outbreak, due to application of public health and social measures to stop the outbreak, precluding determination of some useful subgroup analyses such as VE by comorbidity. Although we were able to adjust for presence of one or more comorbidities, we could not determine VE by any specific comorbidity.”
Round 2
Reviewer 3 Report
In the revised version of the article entitled “Effectiveness of inactivated COVID-19 Vaccines against Delta-variant COVID-19: Evidence from an Outbreak in Inner Mongolia Autonomous Region, China” the authors added my required information. Therefore, I think that this revised version can be published in its current version.